# Treasury Bond Return Data Starting in 1962

**Laurens Swinkels** [1,2] 

1   Erasmus University Rotterdam, Erasmus School of Economics, Business Economics, 3062 PA Rotterdam,
    The Netherlands; lswinkels@ese.eur.nl
2   Robeco Institutional Asset Management, Quantitative Investing, 3014 DA Rotterdam, The Netherlands

**Abstract:** Academics and research analysts in financial economics frequently use returns on government bonds for their empirical analyses. In the United States, government bonds are also called Treasury bonds. The Federal Reserve publishes the yield-to-maturity of Treasury bonds. However, the Treasury bond returns earned by investors are not publicly available. The purpose of this study is to provide these currently not publicly available return series and provide formulas such that these series can easily be updated by researchers. We use standard textbook formulas to convert the yield-to-maturity data to investor returns. The starting date of our series is January 1962, when end-of-month data on the yield-to-maturity become publicly available. We compare our newly created total return series with alternative series that can be purchased. Our return series are very close, suggesting that they are a high-quality public alternative to commercially available data.

**Dataset:** https://doi.org/10.25397/eur.8152748

**Dataset License:** CC-BY

**Keywords:** bonds; government bonds; interest rate; investment; returns; treasury; yield

---

## 1. Summary

It is not straightforward to obtain historical data on government bond returns, unless the data is purchased from commercial data vendors. However, the yield-to-maturity of Treasury bonds with a 10-year maturity is available from the Federal Reserve Economic Data (FRED) database, maintained by the Federal Reserve Bank of St. Louis, with a history going back to January 1962. Investor returns can be approximated with yield-to-maturity data using finance textbook formulas. Our contribution is to transform the publicly available yield-to-maturity time-series into investor returns, and make this data publicly available for academics and research analysts in the finance industry. We show that the returns are very close to alternative Treasury bond return series that have to be purchased from data vendors. Hence, our public Treasury bond return series is of high quality.

Note that time-series of returns are valuable for researchers. Recently, the authors of [1] have published annual data on total returns for equity, housing, bonds, and bills which cover 16 advanced economies from 1870 to 2015. The main advantage of our data is that it is available at the monthly frequency, which is often used in the asset pricing literature. Another example is [2], the authors of which have collected foreign currency sovereign bond prices traded in London and New York since 1815.

## 2. Data Description

The data are available in a spreadsheet with formulas, such that it can be easily updated after publicly available data is added. Here, we describe each of the columns in the spreadsheet in more detail.

- Column A: The last calendar day of each month.
- Column B: The yield-to-maturity of a 10-year fixed maturity Treasury bond. These data are publicly available from [3].
- Column C: The interest rate sensitivity of the price of the 10-year Treasury bond, which is also called the modified duration. This quantity uses as inputs the yield-to-maturity and maturity, using [4] (Equation 4.45 on p. 146).
- Column D: The interest rate sensitivity of the modified duration of the 10-year Treasury bond, which is also called the convexity. This quantity uses as inputs the yield-to-maturity and maturity, using [4] (Equation 4.52 on p. 150).
- Column E: The monthly total return for an investor in the 10-year Treasury bond. This quantity uses as inputs the yield-to-maturity and maturity, derived from [4] (Equation 4.21 on p. 138).
- Column F: The cumulative total return from an investor starting out with $100 on 31 December 1961.
- Column G: Intentionally left blank.
- Column H: The last calendar date of each year.
- Column I: The cumulative return at the last calendar date of each year (from Column F).
- Column J: The annual total return for an investor in the 10-year Treasury bond.

## 3. Methods

We use the daily 10-year Treasury constant maturity rate [3] as our time-series of yield-to-maturity. Since this rate corresponds to a 10-year bond, the remaining maturity is constant: $M_t = 10$. This daily series start on 2 January 1962. In order to calculate a return for January 1962, we assume that the observation for 31 December 1961 equals the observation of 2 January 1962. Note that the monthly series [5] that starts in April 1953 is not useful for our goal, as this is the monthly *average* of daily yields, rather than the end-of-month yield that we require to calculate the monthly return.

Figure 1 contains the time-series of our input data on the 10-year Treasury constant maturity rate. It starts out at about 4 percent, then increases to almost 16 percent in the early 1980s, and has been trending down to around 2 percent.

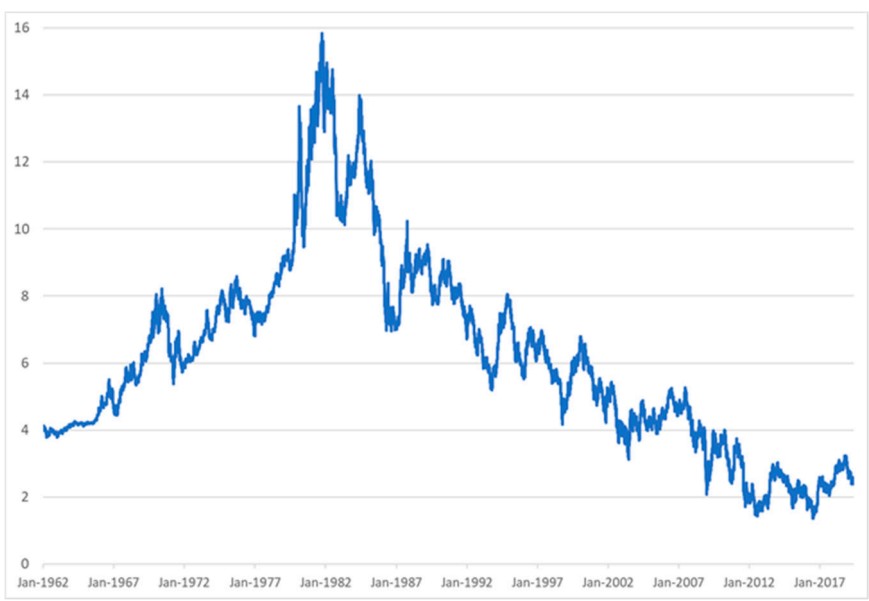

**Figure 1.** The daily Treasury bond 10-year maturity yield-to-maturity (DSG10).

We use [4] for the transformation of yield-to-maturity to investment returns. Note that the formulas that follow are only valid for bonds that have always paid promised cash flows. In case of defaults or debt restructurings, capital losses need to be taken into account separately, and the

following formulas do not adjust for this. Hence, using these formulas for corporate bonds would generally lead to an overestimation of investor returns, as defaults would not be accounted for.

Since the 10-year constant maturity rates that we use as inputs from [3] are based on on-the-run or actively traded securities with maturity close to 10 years, we use the formulas for par bonds. A par bond is a bond that sells at face value, i.e., a bond where the coupon rate equals the bond's yield. This is approximately true for on-the-run bonds.

Given the yield-to-maturity and remaining maturity, the interest rate sensitivity or modified duration of a risk-free bond at par value can be approximated by

$$D_t(Y_t, M_t) = 1/Y_t \cdot [1 - 1/(1 + 0.5 \cdot Y_t)^{2 \cdot M_t}], \tag{1}$$

with $Y_t$ the yield-to-maturity at time $t$ and $M_t$ the remaining maturity of the bond at time $t$. The convexity $C_t$ of a par bond, which measures the non-linear relationship between the price and yield of a bond can be approximated with

$$C_t(Y_t, M_t) = 2/(Y_t^2) \cdot [1 - 1/(1 + 0.5 \cdot Y_t)^{2 \cdot Mt}] - (2 \cdot M_t)/[Y_t \cdot (1 + 0.5 \cdot Y_t)]^{2 \cdot M_t + 1}. \tag{2}$$

For better approximations, the dispersion in maturity of the bond portfolio is needed as an input, which we do not have available. Now, we can calculate the investment return over period $t$ when we know the yield at the beginning and the end of the period, and the maturity of the bond

$$R_t(Y_{t-1}, Y_t, M_t) = Y_{t-1} - D_t \cdot (Y_t - Y_{t-1}) + \frac{1}{2} \cdot C_t \cdot (Y_t - Y_{t-1})^2 \tag{3}$$

where the yield in the first term on the right-hand side of the equation needs to be expressed as a percentage per period that the return is measured (i.e., $(1 + Y_t)^{1/12} - 1$ of the annual yield when calculating a monthly return), while the duration and yields in both other terms on the right-hand side should be measured in the same units (e.g., per annum). This equation shows that if the interest rate does not change, the two terms on the right equal zero and the return is the yield at the beginning of the period. Increases in the yield will reduce the return, but this decrease is smaller for bonds with higher convexity.

## 4. User Notes

We have access to several alternative data series on U.S. bond returns that are available for purchase or subscription. We compare our approximation to these commercially available data series. The comparison is not exhaustive, as we do not have access to all possible commercial sources, and for the existing sources not for the entire sample period.

- Global Financial Data (GFD)
  We have available the U.S Treasury total return index (index id: TRUSG10M) for the start of the sample until March 2017 from [6]. Over this period, they state that they based their data on that from the Federal Reserve.
- Center for Research in Securities Prices (CRSP)
  Through the Wharton Data Research Service, we have available the 10-year bond returns series prepared by CRSP in their "US Treasury and Inflation Indexes" service [7]. This series is available through our entire sample period.
- Ibbotson
  We have access to [8], which contains intermediate and long-term government bond returns at the monthly frequency from the start of the sample to the end of 2005. This is an annual update of [9]. In order to obtain a series that is similar to a series with a 10-year maturity, we took a portfolio of both series with weight 63.5% in the long-term and 36.5% in the intermediate-term government bond. The volatility of both series is equal for this choice. However, since the maturities of the

intermediate-term and long-term government bond series may change over time, this series does not represent a 10-year maturity at each point in time.

- Bloomberg

The U.S. intermediate-term (index id: 49) and long-term (index id: 50) government bond return series of Bloomberg Barclays [10] are available at the monthly frequency from January 1973 to 2018. In order to obtain a series that is similar to a series with a 10-year maturity, we took a portfolio of both series with weight 68.4% in the long-term and 31.6% in the intermediate-term government bond. The volatility of both series is equal for this choice. However, since the maturities of the intermediate-term and long-term government bond series may change over time, this series does not represent a 10-year maturity at each point in time.

Table 1 displays the annualized average returns, standard deviations (i.e., volatility) of the returns, the minimum monthly return, and the maximum monthly return. It also contains the intercept, slope, and r-squared of the regression equation where the dependent variable is our series and the explanatory variable is the alternative series. We also display the p-value of the Wald-test statistic of the joint null hypothesis that the intercept is zero and the slope equals one, which would mean that both series are equal to each other. Since the data is available over different periods, we separate January 1962 to December 2018, January 1962 to December 2005 (Ibbotson), and January 1973 to December 2018 (Bloomberg Barclays). Note that the GFD series ends in March 2017, but we do not separate this out.

**Table 1.** Comparison of return characteristics for different data sources.

| Sample | Statistic | Our Series | GFD | CRSP | Ibbotson | Bloomberg |
|---|---|---|---|---|---|---|
| 1962–2018 | Average | 6.76% | 6.80% | 6.63% | | |
| | Volatility | 7.86% | 7.91% | 7.54% | | |
| | Minimum | −7.41% | −7.91% | −6.68% | | |
| | Maximum | 13.23% | 12.69% | 10.00% | | |
| | Intercept | | 0.01% | 0.01% | | |
| | Slope | | 1.001 | 1.008 | | |
| | R-squared | | 0.996 | 0.934 | | |
| | Wald p-val | | 0.143 | 0.344 | | |
| 1962–2005 | Average | 7.46% | 7.34% | 7.28% | 7.48% | |
| | Volatility | 8.02% | 7.99% | 7.80% | 8.02% | |
| | Minimum | −7.41% | −7.91% | −6.68% | −7.40% | |
| | Maximum | 13.23% | 12.69% | 10.00% | 14.04% | |
| | Intercept | | 0.01% | 0.02% | 0.02% | |
| | Slope | | 1.003 | 0.990 | 0.968 | |
| | R-squared | | 0.998 | 0.926 | 0.936 | |
| | Wald p-val | | 0.031 | 0.618 | 0.0164 | |
| 1973–2018 | Average | 7.46% | 7.53% | 7.26% | | 7.74% |
| | Volatility | 8.24% | 8.30% | 7.74% | | 8.24% |
| | Minimum | −7.41% | −7.91% | −6.68% | | −6.81% |
| | Maximum | 13.23% | 12.69% | 10.00% | | 12.51% |
| | Intercept | | 0.01% | 0.00% | | −0.01% |
| | Slope | | 1.002 | 1.032 | | 0.972 |
| | R-squared | | 0.995 | 0.934 | | 0.945 |
| | Wald p-val | | 0.109 | 0.015 | | 0.012 |

It can be seen that our series is a close match to each of the alternatives, as average returns and risk characteristics are similar. The intercepts of each of the regressions close to zero, and the slope coefficients close to one, with r-squared values above 0.92. Note, however, that the p-values are below 0.05 in several cases, which means that they statistically reject that the series are equal. This is due to the large r-squares, which leads to extremely narrow confidence bounds around the slope coefficient. Hence, even though the coefficients jointly are statistically significantly different from zero

and one, they are not economically significantly different. Clearly, the GFD series that is often used in academic research (see, for example, [11–14]) is extremely close to ours with an r-square of 0.996 over the entire sample period from 1962 to 2018. The closeness of the series is also illustrated in Figure 2, in which we plot the cumulative value of a $100 investment in the Treasury bond at 31 December 1961. We see that the data series are so close that for many periods the lines overlap. Therefore, researchers without access to commercial data providers may use our data series and obtain very similar results to researchers with access.

This data series has a few limitations. First, it covers only U.S. data going back to 1962. Our method could be easily extended to convert other historical government bond yield-to-maturity data to investor returns, but we do not have these publicly available. Second, our return approximations are based on formulas valid for par bonds. Although our source data on yield-to-maturity tend to use on-the-run bonds that are close to par, they need not be over the entire sample period, slightly reducing the accuracy of the approximation. Note that the yield-to-maturities that we use may be estimates from liquid bond prices with close but not exactly equal to 10 years maturity. Therefore, our time-series data with investor returns are hypothetical, as a Treasury bond with exactly 10 years to maturity was not available for investors each month-end during our sample. Finally, our data series represent the return on a Treasury with 10 years maturity. While this is appropriate for many applications, other applications may require returns on the entire bond market, in which each outstanding bond is weighted by its market capitalization [15].

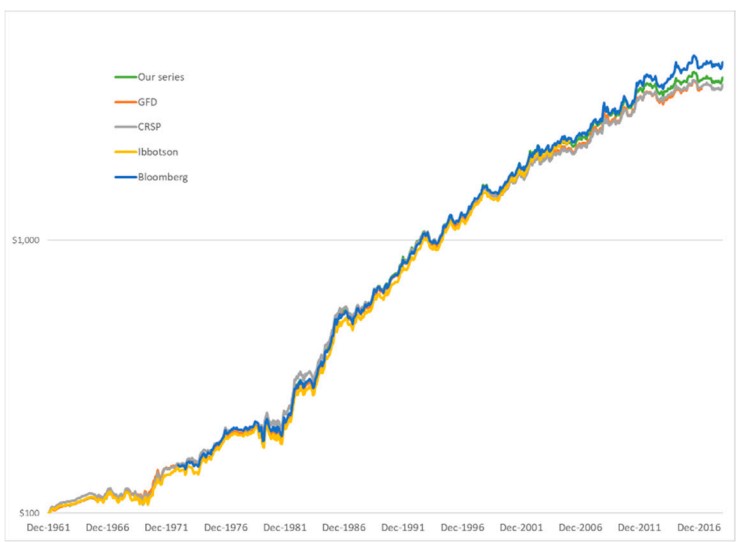

**Figure 2.** Cumulative returns of our and alternative data series.

**Funding:** This research received no external funding.

**Acknowledgments:** I would like to thank Guido Baltussen, Trevin Lam, and Pim van Vliet for helpful comments.

**Conflicts of Interest:** The author declares no conflict of interest. The views expressed in this paper are not necessarily shared by Robeco Institutional Asset Management.

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
