# Peer review of "Treasury Bond Return Data Starting in 1962"

_data, 1962_

Reviewer 1 Report

Please see my report.

Author Response

I would like to thank the reviewer for his positive feedback on the added value of this "data descriptor" and the valuable suggestions to further enhance the manuscript.  

1. I have added the requested regression parameters for the intercept and slope, as well as the R-squared and the p-values of the Wald-test for the joint hypothesis that the intercept equals zero and the slope equals one. As you can see, the p-values are below 0.05 for a few cases. Due to the extremely good fit, the residual variance is tiny, and slight deviations are statistically significant, even when the intercept is extremely close to zero and the slope extremely close to one. I include this information because it is important to judge the closeness, I perfer to emphasize the high R-squared rather than the rejection of the null hypothesis.

2. Thank you for this valuable suggestion. I have now included a comment on page 3 (ln 74):

Since the 10-year constant maturity rates that we use as inputs from [3] are based on on-the-run or actively traded securities with maturity close to 10 years, we use the formulas for par bonds. A par bond is a bond that sells at face value, i.e. a bonds where the coupon rate equals the bond’s yield. This is approximately true for on-the-run bonds.

3. I have changed the equation number to (3). Thank you for pointing this out.

4. I have now added the equation numbers on pages 1 and 2:

Column C: The interest rate sensitivity of the price of the 10-year Treasury bond, which is also called the modified duration. This quantity uses as inputs the yield-to-maturity and maturity, using [2] (Equation 4.45 on p. 146).

Column D: The interest rate sensitivity of the modified duration of the 10-year Treasury bond, which is also called the convexity. This quantity uses as inputs the yield-to-maturity and maturity, using [2] (Equation 4.52 on p. 150).

Column E: The monthly total return for an investor in the 10-year Treasury bond. This quantity uses as inputs the yield-to-maturity and maturity, using [2] (Equation 4.21 on p. 138).

Reviewer 2 Report

1.     The abstract should be improved and have to provide more structured aim, scope and background, to state the principal objectives and scope of the investigation, to describe the methods employed, to summarize the results, to state the principal conclusions, the originality and value of the paper.

2.     In the introduction, context of the research should be established, the purpose and/or hypothesis that was investigated should be stated. The main idea, importance, novelty, etc. can be indicated in this section.

3.     The main text/ literature review should include previous research on the subject, trace the intellectual progression of the field. The interpretation of previous research should be given. Theoretical study should be based on recently published papers from high level scientific journals indexed in Web of Science database (years 2017-2019).

4.     Please check all references, they all must be mentioned in the paper. Data sources (GFD, GRSP, Bloomberg) should be mentioned in the list of references.

5.     Comparisons with other studies have to be provided in the discussion section.

6.     Please provide limitations of your research.

7.     The conclusions must concisely summarize the main points of the paper.

Author Response

I would like to thank the reviewer for his/her valuable feedback to this "Data Descriptor" manuscript.

1. I have rewritten the abstract as per the journal guidelines: 

Abstract: The abstract should be a total of about 200 words maximum. The abstract should be a single paragraph and should follow the style of structured abstracts, but without headings: 1) Background: Place the question addressed in a broad context and highlight the purpose of the study; 2) Methods: Describe briefly the main methods or treatments applied. Include any relevant preregistration numbers, and species and strains of any animals used. 3) Results: Summarize the article's main findings; and 4) Conclusion: Indicate the main conclusions or interpretations. The abstract should be an objective representation of the article: it must not contain results which are not presented and substantiated in the main text and should not exaggerate the main conclusions.

2. I follow the journal guidelines for "Data Descriptors", outlined as follows on the journal's website:

Data Descriptors comprise the following sections:

Summary: A short summary of the dataset, methods, background information on why and how the dataset was collected, short description of funded or unfunded research projects that are or will eventually be based on the dataset, and list of publications based on the dataset that were possibly already published. Optionally, authors may wish to describe potential benefits of publicly releasing and describing the dataset. In general, the Summary section is similar to an introduction section in a research article.

Data Description: What data is contained? Which format? How can it be read and interpreted? For example, in tabular data give a full description of each column heading.

Methods: Main methods applied to collect and treat, as well as to use and reuse the data. Notes on validation and curation techniques applied. Notes on data quality, noise, etc.

Usage notes (optional): Further notes on the usage of the dataset that will help other researchers to access and further understand practical aspects of working with the data. If there are ethical or compelling commercial reasons that the data cannot be made available, either in part or in full, these should be described in as much detail as possible. You should make clear how the data can be accessed and if there are circumstances in which access would be denied (e.g. if complying with the request would compromise anonymity of human participants or if an embargo applies); we recommend full access wherever possible.

These guidelines do not include an introduction or literature review section.

3. I follow the journal guidelines for "Data Descriptors" as outlined above in point 2. However, I based on the suggestion of the reviewer, I have included a reference to Jorda et al. (2019), which contains new financial markets data at the annual frequency and Meyer et al. (2019) who document foreign-currency government bond returns since 1815. I thank the reviewer for this suggestion.

4. Thank you for this suggestion. Now all data sources have been referenced.

5. For a data descriptor, the comparison with other data series seems to be the only comparison that is of value to the user of the data. I have included additional references to recent top articles published using the alternative commercially available Global Financial Data.

6. As requested by the reviewer, I have added a paragraph on limitations of the constructed data series.

7. The journal guidelines specify that there is no conclusion section for "Data Descriptors"  

Round  2

Reviewer 1 Report

All my comments have been properly addressed and I have no further questions. Therefore I recommend accepting the paper in present form.

Reviewer 2 Report

The article can be accepted for publication.